# The lncRNAs in HBV-Related HCCs: Targeting Chromatin Dynamics and Beyond

**DOI:** 10.3390/cancers13133115

**Published:** 2021-06-22

**Authors:** Vincenzo Alfano, Mirjam B. Zeisel, Massimo Levrero, Francesca Guerrieri

**Affiliations:** 1Cancer Research Center of Lyon (CRCL), UMR Inserm 1052 CNRS 5286 Mixte CLB, Université de Lyon 1 (UCBL1), 69003 Lyon, France; vincenzo.alfano@inserm.fr (V.A.); mirjam.zeisel@inserm.fr (M.B.Z.); 2Hospices Civils de Lyon, Hôpital Croix Rousse, Service d’Hépato-Gastroentérologie, 69004 Lyon, France; 3Department of Medicine SCIAC, University of Rome La Sapienza, 00161 Rome, Italy

**Keywords:** HBV, HCC, LncRNAs, epigenetic regulation, biomarkers

## Abstract

**Simple Summary:**

Hepatocellular carcinoma (HCC), a common and fast rising cause of cancer, is responsible for over 800,000 deaths/year. Chronic hepatitis B virus (HBV) infection accounts for >50% of the cases worldwide. Long non-coding RNAs (lncRNAs), untranslated transcripts longer than 200 nucleotides, by acting both in the nuclear and cytoplasmic compartments, regulate gene expression both at the transcriptional and post-transcriptional levels. The lncRNAs have been involved in the development and progression of many cancers, including HCC. In this review, we describe the role of lncRNAs in HBV infection and HBV-related liver carcinogenesis and discuss the potential of lncRNAs as predictive or diagnostic biomarkers.

**Abstract:**

Hepatocellular carcinoma (HCC) represents the fourth leading and fastest rising cause of cancer death (841,000 new cases and 782,000 deaths annually), and hepatitis B (HBV), with 250 million people chronically infected at risk of developing HCC, accounts for >50% of the cases worldwide. Long non-coding RNAs (lncRNAs), untranslated transcripts longer than 200 nucleotides, are implicated in gene regulation at the transcriptional and post-transcriptional levels, exerting their activities both in the nuclear and cytoplasmic compartments. Thanks to high-throughput sequencing techniques, several lncRNAs have been shown to favor the establishment of chronic HBV infection, to change the host transcriptome to establish a pro-carcinogenic environment, and to directly participate in HCC development and progression. In this review, we summarize current knowledge on the role of lncRNAs in HBV infection and HBV-related liver carcinogenesis and discuss the potential of lncRNAs as predictive or diagnostic biomarkers.

## 1. Introduction

Hepatitis B virus (HBV) infection is a global health problem that accounts for 257 million chronically infected people, and it is considered the major risk factor for hepatocellular carcinoma (HCC). The genetic landscape of HCCs [1] is highly heterogeneous harboring multiple mutations with significant incidence in HCC-affected patients. Various mechanisms support a direct involvement of HBV in driving hepatocarcinogenesis [2] and the two best described are (a) the HBV integration into the host genome [3] and (b) the cancerogenic role of the multifunctional HBV protein X (HBx) [4,5] and the aberrant transactivation of host genes [6]. 

Important players involved in the molecular mechanisms responsible for the development of cancer are the long non-coding RNAs (lcnRNAs) [7], which are RNA transcripts longer than 200 nucleotides that are not translated into proteins. They are implicated in gene regulation at the transcriptional and post-transcriptional levels and are now at the forefront of studies exploring the molecular pathogenesis and the diagnosis of cancer [8]. Indeed, lncRNAs can be found in different cellular compartments as well as in body fluids, making them attractive biomarkers. Recent evidence has shown that HBV is able to induce the expression of several host long non-coding RNAs [6,9]. However, in contrast to other viruses, HBV-derived lncRNAs have not yet been described [10]. Upon viral infections, lncRNAs can exert different effects, either by inhibiting virus replication or by promoting viral spread. Indeed, lncRNAs can modulate the host’s immune responses by stimulating or suppressing antiviral mechanisms, or they can directly promote viral replication at different steps of the viral life cycle (DNA transcription or RNA/protein/particle stability) [10]. Thus, the interplay between HBV and host lncRNAs could contribute to regulate the different steps leading to HCC in chronically HBV-infected patients. Thanks to high-throughput sequencing techniques and machine learning analysis, hundreds of lncrRNAs have been associated to HBV infection as well as to HBV-related HCC, and they have been shown to exert different functions at different steps (infection, cancerogenesis, tumor progression, or drug resistance) by acting in different cellular compartments (nucleus/ cytoplasm) [11]. Altogether, lncRNAs hold promise both as biomarkers for HCC diagnosis and as targets for HCC treatment (Figure 1). 

In this review, we focus on those lncRNAs that have been described to be modulated in HBV infection and have been found to be dysregulated in HBV-related HCC in order to provide an overview of the mechanisms underlying the involvement of lncRNAs in HBV-related HCC.

## 2. Multiple Zipcodes Direct the Subcellular Localization and Functions of lncRNAs

The study of lncRNAs has been challenging due to their low expression and the lack of tools to assess their subcellular localization and functions. Inside cells, lncRNAs have been detected both in nuclear and cytoplasmic compartments. Recently, two valuable tools have been developed to help study the subcellular localization and its impact on lncRNA functions: lncATLAS is a comprehensive resource of lncRNA localization in 15 human cell lines based on RNA-sequencing datasets [12], and *Locate-R* is a novel method for predicting the subcellular location of lncRNAs based on their nucleotide sequence [13]. Noteworthily, in human cells, lncRNAs tend to be more often nuclear [14]. The localization of lncRNAs reflects their specific function (Figure 1), and in recent years, as for miRNAs, several zipcodes, triggering lncRNA cellular distribution have been identified. The localization of lncRNAs is defined not only by the type of molecule with which they interact (protein, DNA or RNA) but also by conserved sequence motifs, transposable elements (TEs), or post-transcriptional modifications within the lncRNAs [15,16]. An additional mechanism that contributes to the enrichment of lncRNAs in the nucleus is the greater rate of lncRNA degradation in the cytoplasm [15,17].

## 3. Nuclear lncRNAs

### 3.1. lncRNAs Regulate the Recruitment and Function of Chromatin-Modifying Complexes

Nuclear lncRNAs cooperate with protein complexes that modulate nuclear organization, compartment formation, phase separation, and epigenetic regulation [18]. Given their ability to bind both DNA and proteins, nuclear lncRNAs have been described to contribute to chromatin remodeling and to the modulation of gene expression in various manners. Accumulating evidence indicates that lncRNAs can alter the chromatin status by sequestering (decoy model), guiding toward their target loci (guide model), or tethering (scaffold model) transcription factors or chromatin-modifying complexes. By recruiting chromatin-modifying complexes, such as polycomb repressor complex (PRC) 1, PRC2, and LSD1 on specific genomic loci, several lncRNAs have been shown to play an important role in the modulation of gene expression by inhibiting or activating transcription [19,20] (Figure 2). 

The best documented example of chromatin modifying complexes interacting with lncRNAs is PRC2. Notably, ~20% of all human lncRNAs have been shown to associate with PRC2 [21] to facilitate its recruitment to chromatin and to modulate its functions. PRC2 is responsible for imposing repressive methyl groups at lysine 27 of the histone H3 tail (H3K27me3) via its catalytic subunit Enhancer of Zeste homolog 2 (EZH2). EZH2 is overexpressed in HCC [22], and it has been proposed as a histologic immunomarker for the diagnosis of early HCCs [23]. In addition to PRC2-mediated gene repression, non-canonical functions that do not involve histone methylation and lead to transcriptional activation have also been described for EZH2 [24]. Additional lines of evidence suggested that lncRNAs can tune PRC2 activity without affecting its recruitment [25,26,27]. Recent data indicate that EZH2 binding to RNA is essential for the modulation of its catalytic activity [19]. Direct EZH2–RNA binding has been demonstrated using EZH2–RNA immunoprecipitation followed by next-generation sequencing (RIP-Seq) [26] and by in vitro binding experiments [19,25]. RNA–PRC2 binding depends on the RNA length, sequence, and structure [19,25,27]. Interestingly, PRC2 displays a high affinity for G-rich RNAs, especially when they form G-quadruplex structures [28]. The inhibition of EZH2 methyltransferase activity toward histone and non-histone substrates by RNA [29] provides a possible mechanistic explanation for the non-canonical functions of EZH2. However, this does not exclude that some lncRNAs might promote PRC2 recruitment to target genes [30]. 

The role of lncRNAs in modulating other chromatin-modifying complexes and their potential role in HCC are less established. However, a recent study has reported the association between hundreds of lncRNAs and specific histone modifications including H3K27me3, H3K4me1/3, and H3K9me3, suggesting that additional chromatin-modifying complexes could interact with lncRNAs in a cell-type specific manner [31].

### 3.2. Nuclear lncRNAs in HBV-Associated HCC

Given the large number of lncRNAs found to be deregulated in HBV-related HCC patients, we will describe below only lncRNAs for which the epigenetic mechanism of action has been investigated and focus on lncRNAs that have been directly correlated to HBV infection (HEIH, HOTAIR, TUC338, HOTTIP, DLEU2, MVIH, PVT1, ANRIL) or HBx expression (UCA1, LINC00152, MALAT1) and have been shown to affect HBV infection or promote hepatocarcinogenesis or both.

HEIH (High Expression in HCC) is one of the first lncRNAs that was demonstrated to be upregulated in HBV-related HCC patients [32]. By providing the first link between oncogenic lncRNAs and chromatin-modifying complexes, and in particular EZH2, in HCC, this study not only uncovered a novel HCC biomarker but also opened the door to the exploration of lncRNA functions in hepatocarcinogenesis. Another study subsequently highlighted the overexpression of HEIH in HBV-related HCC tissues [33], and we have recently demonstrated that HEIH is upregulated following HBV infection of primary human hepatocytes (PHHs) [34].

The first example of an lncRNA that is hijacked by HBV to promote its replication and is associated with poor prognosis of liver cancer in chronically infected HBV patients was the lncRNA HOTAIR (HOX transcript antisense RNA) [35]. The expression of HBx in HBV-infected cells activates PLK1, which in turn phosphorylates SUZ12, a subunit of PRC2, in G2 phase [36]. This results in the downregulation, by a mechanism that remains to be clarified, of the DDX5 helicase bound to HOTAIR, the preferential association of HOTAIR with the E3-ligase Mex3b, and the ubiquitination of SUZ12, leading to its proteasomal degradation [37] (Figure 2). As a result, the transcriptional repression exerted by the DDX5/PRC2/HOTAIR complex is impaired, leading to the transcriptional reactivation of PRC2/HOTAIR target genes, including EpCAM and other cellular genes expressed in hepatic cancer stem cells (hCSC) [37], as well as to an increase in HBV cccDNA transcription and HBV replication [37]. While the molecular mechanisms underlying HOTAIR-PRC2 interaction in vivo are still debated, a direct interaction of the 5’ domain of HOTAIR with EZH2 has been demonstrated in vitro [20], and it has been shown that the interaction involves the repetitive guanine stretches (G-tracts) found in the HOTAIR D1 helix [38].

Another lncRNA positively correlated with HBV-related HCC and induced in HBx-expressing cells is UCA1 (Urothelial Cancer Associated 1) [39]. By directly binding/recruiting EZH2 in the nucleus (Figure 2), UCA1 represses the expression of p27, which regulates the activity of CDK2. The knockdown of UCA1 in hepatoma cells decreased the binding of EZH2 and H3K27me3 on the p27 promoter and significantly suppressed tumor growth in a xenograft mouse model [39]. Thus, by increasing UCA1, HBx can contribute to CDK2-mediated proliferation of hepatoma cells [39]. It remains to be determined whether UCA1 expression also impacts on HBV replication.

LINC00152 has been described to repress gene expression by interacting with EZH2 [40] as well as to activate the transcription of EpCAM by binding to its promoter, thereby favoring cell proliferation in hepatoma cells [41]. LINC00152 was found to be hypomethylated during hepatocarcinogenesis [42]. HBx has been shown to increase LINC00152 in hepatoma cells, and higher LINC00152 levels have been associated with HBx expression in HBV-related HCC tissues as well as with poor prognosis in HCC patients [40]. While HBx contributes to LINC00152 regulation in HBV-infected liver cells, this lncRNA has not been demonstrated to modulate HBV replication.

TUC338 is another lncRNA induced by HBV [34] and upregulated in HCC [43]. This DNA-binding RNA recognizing p53- and Pax6-binding motifs on the genome has been proposed to act as a scaffold for the interaction between specific binding proteins and as a guide to route transcription factors to specific genomic loci [44]. The siRNA-mediated silencing of TUC338 in HCC cells altered the expression of 611 genes involved in biological processes related to cell proliferation. The increased expression of TUC338 in HCC could indeed prevent p53 from DNA binding at these sites and thereby functionally silence these p53 target genes [44]. These data suggest that TUC338 may contribute to tumor growth in HCC via sequence-defined cis-binding sites to modulate the expression of genes involved in aberrant cell proliferation.

The transcription of lncRNA MALAT1 (Metastasis-Associated Lung Adenocarcinoma Transcript 1) is also induced by HBx [34], and numerous studies have shown that MALAT1 promotes HCC progression, metastasis, and recurrence after liver transplantation [45]. While its secondary structure, comprising a bipartite triple helix, has been extensively studied [46], the compartmentalization and the mode of action of MALAT1 are still to be investigated. In addition to the many cytoplasmic functions attributed to MALAT1, this lncRNA has been shown to be also enriched at the periphery of nuclear speckles and to bind to SR proteins, which are splicing factors that are required for pre-mRNA alternative splicing [47]. MALAT1-depleted cell extracts showed a significant increase in the levels of non-phosphorylated SR proteins. By interacting with a specific set of SR splicing factors, MALAT1 acts as a ‘‘molecular sponge’’ to regulate SR protein activity. Interestingly, small molecules specifically targeting the MALAT1 triple helix structure have been identified, and they could represent novel strategies for the treatment of MALAT1-driven cancer and investigate its functions [48]. 

We have recently shown that DLEU2 (Deleted in Lymphocytic Leukemia 2) is activated by HBV infection in PHHs and deregulated in HCC [34]. DLEU2 directly binds HBx and EZH2 and in silico modeling suggested that, depending on the relative abundance of EZH2 and HBx, they compete for partially overlapping sites on DLEU2, thereby modulating EZH2/PRC2 functions [34] (Figure 2). By interacting with DLEU2, HBx modulates the transcription of several host genes and the HBV cccDNA minichromosome. Indeed, in HBV-infected cells, HBx–DLEU2 association at target host promoters and on the HBV cccDNA relieves EZH2-mediated repression and leads to the transcriptional activation of a subset of EZH2/PRC2 host target genes as well as to HBV replication [34]. Of note, DLEU2–EZH2–HBx-mediated changes in chromatin state did not affect PRC2 promoter occupancy at the promoter of the cellular target genes [34]. 

PVT1 and ANRIL, two other EZH2-associated lncRNAs upregulated in HBV-related HCC tissues, have been demonstrated to also interact with other PRC2 subunits [49,50]. PVT1 has been shown to interact with EZH2, EED, and SUZ1 [49] (Figure 2). This PVT1-PRC2 interaction leads to a loss of EZH2 recruitment to the c-Myc oncogene promoter and the concomitant reduction of the H3K27me3 histone mark. Interestingly, PVT1 is also an oncofetal RNA (lncRNA-mPvt1), and its expression is associated with stem cell-like properties in a murine model [51]. hPVT1 promotes cell proliferation, cell cycling, and the acquisition of stem cell-like properties in HCC cells by directly binding the protein NOP2. Furthermore, the interaction of ANRIL, the lncRNA that displays the highest expression in HBV-related HCC tissues [50], with both EZH2 and SUZ12 guides PRC2 to the Kruppel-like factor 2 (KLF2) promoter, leading to an increase of H3K27 tri-methylation as well as decreased transcription and cell proliferation [52]. 

In contrast to the above-mentioned lncRNAs, the lncRNA MVIH is upregulated in PHHs upon HBV infection [34] and in HCC samples from a large cohort of HCC patients [53] but does not physically associate with EZH2 [53]. Instead, it has been shown to associates with Phosphoglycerate kinase 1 (PGK1), which is a protein inhibiting angiogenesis [53]. By sequestering PGK1 and reducing its secretion, MVIH could contribute to activate angiogenesis, which is an early event in HCC development. MVIH overexpression was also correlated with HCC recurrence after hepatectomy, suggesting that MVIH might be an attractive prognostic biomarker to predict HCC recurrence after hepatectomy [53]. 

Finally, a recent study has identified HOTTIP (HOXA transcript at the distal tip) as an lncRNA highly expressed in HBV-positive HCCs that is capable of suppressing HBV replication as well as decreasing hepatitis B viral surface antigen (HBsAg) and hepatitis B viral e antigen (HBe) production [54]. Mechanistically, the DNA polymerase of the virus promotes CREB1 expression, which controls HOTTIP transcription in an HBx-independent manner. In turn, HOTTIP upregulates the transcription factor HOXA13 that binds to and suppresses the activity of the HBV promoter Enh I/Xp, resulting in a decrease in virus replication [54]. Interestingly, HOTTIP and HOXA13 have been previously associated with HCC [55], and HOXA13 has been proposed to play an important role in the host immune responses to influenza virus infection [56]. HOTTIP has been shown to bind EZH2 in renal carcinoma [57], but EZH2–HOTTIP interactions have not yet been reported in the liver.

## 4. Cytoplasmic lncRNAs

In the last decade, numerous lncRNA-mediated post-transcriptional mechanisms of gene regulation have been shown to take place in the cytoplasm: (a) direct binding to complementary sequences on miRNAs and de-repression of miRNAs targets [58] (HULC, HBx-LINE1); (b) binding to cytoplasmic proteins and affecting their stability (HUR1, DREH); (c) acting as precursors of miRNAs (Ftx, H19); and (d) coding for small peptides through small open reading frames (smORFs) [59] (Figure 1).

### 4.1. ceRNAs (Competing Endogenous RNA) Directly Bind miRNAs

The lncRNAs referred to as Competing Endogenous RNA (ceRNAs) exert a sponge activity by sequestering complementary miRNAs. The prototype of this class of lncRNAs is the oncogenic lncRNA HULC (Highly Upregulated in Liver Cancer). HULC was the first lncRNA identified to be specifically upregulated in HCC [60]. A subsequent elegant study showed that the CREB-dependent expression of HULC is part of an auto-regulatory loop in which CREB has an inhibitory effect on the expression of miR-372, leading to the upregulation of this lncRNA in HCC [61]. HBx binds the HULC promoter region via CREB and upregulates this lncRNA in liver cells and hepatoma cells [62]. HULC is known to bind nine different miRNAs [58], and by acting as a ceRNA, HULC has been suggested to modulate the angiogenesis, proliferation, migration, invasion, and metastasis of HCC cells, thereby contributing to the development of HCC [63]. Furthermore, HULC has recently been associated with the sensitivity of HCC cells to chemotherapeutic agents [64]. By suppressing miR-6825-5p, miR-6845-5p, and miR-6886-3p, HULC de-represses USP22, which in turn stabilizes Sirt1, leading to HCC cell autophagy and oxaliplatin resistance [64]. 

Another intriguing example of this class of lncRNAs is the viral chimeric fusion transcript HBx-LINE1 [65]. The integration of HBV into a normally silenced region of chromosome 8p11.21 generates this RNA, whose relative abundance serves as a prognostic parameter in HBV-related HCC patients [65]. By sequestering miR-122, this ceRNA has been shown to activate the Wnt signaling pathway and to promotes hepatic injury [66].

### 4.2. LncRNAs Altering Protein Stability

Several lncRNAs have been shown to be modulated by HBx and to contribute to HCC by affecting the stability of defined proteins involved in carcinogenesis [67]. HUR1 has been shown to be highly expressed in the HBV transgenic cell line HepG2-4D14 and to be upregulated by HBx in transfected HepG2 cells [67]. HUR1 directly binds the tumor suppressor p53 and inhibits its recruitment on the p21 and Bax promoters in HepG2 cells. Furthermore, HUR1 promotes DEN-induced HCC development in vivo in transgenic mice [67]. 

In contrast to HUR1, the lncRNA DREH (Downregulated expression by HBx) is repressed by HBx and acts as a tumor suppressor in the development of HBV-related HCC [68]. By interacting with vimentin, DREH has been shown to decrease its levels, thereby changing the structure of the cytoskeleton [68]. Low DREH expression correlates with poor survival in HCC patients [68,69]. In line with its anti-tumor activity, DREH exogenous expression in Hepa1-6 cells reduces both the number of tumors and extrahepatic metastasis in nude mice grafts.

Finally, DANCR (differentiation antagonizing non-protein coding RNA) is an lncRNA upregulated in HBV-related HCCs [70] that has been associated to the stemness features in HCC cells [71]. DANCR knockdown leads to the inhibition of HCC growth and metastasis in vivo in an HCC xenograft mouse model and β-catenin signaling in HCC cells [70]. In HCC cells, DANCR has also been reported to positively regulate the expression of HNRNPA1 [72], which is a protein that is upregulated in HCC and associated with poor survival [73]. Interestingly, DANCR has been shown to bind EZH2 and to affect its stability, thereby affecting osteosarcoma cancer growth, invasion, and metastasis [74]. Whether this also applies to HCC cells remains to be determined.

### 4.3. LncRNA Precursors of miRNAs

Several lncRNAs harboring embedded miRNA sequences have been suggested to contribute to HBV-related HCC. Ftx, an lncRNA that is upregulated in HBV-related HCCs, has been shown to contain two miRNA clusters, namely miR-374b/421 and miR-545/374a, in its introns. Interestingly, the miR-545/374a cluster has been also shown to be upregulated in HBV-related HCCs and to significantly correlate with prognosis-related clinical features, including histological grade, metastasis, and tumor capsule [75]. 

Another example is lncRNA H19, which is encoded by the complex H19 locus that leads to various transcripts: the main transcript lncRNA H19, the embedded miR-675, and the two antisense transcripts 91H and H19 opposite tumor suppressor (HOTS). These different transcripts may explain the opposite functions that have been attributed to H19 [76]. As the precursor for miR-675, H19 has been reported to be significantly upregulated in patients with chronic hepatitis B [77] and in HBV-infected PHHs [34]. It has been shown that lncRNA H19/miR-675 modulates in HBx-transfected cells ATP concentration, glucose consumption, and lactate levels [77]. Furthermore, H19 expression has also been negatively related to sorafenib sensitivity in HCC cells [78]. In line with the debated role of H19 in tumorigenesis, conflicting reports about its role in HCC in vivo have been published [79,80].

### 4.4. LncRNAs Encoding Small Peptides

Recent evidence has suggested that some lncRNAs can have biological functions through encoded peptides [81,82]. Indeed, several lncRNAs have been shown to contain a smORF encoding for a small peptide (less than 100 amino acids) [83]. A peptidomic approach to detect short ORF (sORF)-encoded polypeptides (SEPs) has revealed that only a small fraction of lncRNAs are translated [59]. Although the functions of these micropeptides are still elusive, growing evidence implicates them in cancer progression. LINC00998-encoded 59 amino acid (aa) peptide SMIM30 is expressed in liver-derived cells and in HCC tissues [84], and high levels of SMIM30 correlate with poor survival in HCC patients [84]. SMIM30 promotes HCC progression by interacting with the tyrosine kinases SRC/YES1 to activate the mitogen-activated protein kinase (MAPK) pathway, thereby modulating cell proliferation and migration [84].

## 5. Extracellular lnRNAs (ex-lncRNAs)

Coding and non-coding RNAs, including miRNAs, lncRNAs, and circular RNAs (circRNAs), can be detected in extracellular fluids. A meta-analysis of publications on lncRNAs in HCC has shown that increased levels of several extracellular lncRNAs (ex-lncRNas) were associated with poor prognosis in HCC patients, despite the strong heterogeneity among the incorporated studies [85]. To date, there are no standardized methods to detect ex-lncRNAs, and current small RNA-seq protocols often fail to detect all RNA species in plasma. Indeed, a recent study has speculated that RNases in human blood could degrade RNA dinucleotide bonds, leaving a 5’-OH and 3’-P product that is unsuitable for standard ligation-based library preparation protocols and suggested to include a step with T4 polynucleotide kinase prior to library preparation to improve the accuracy of ex-lnRNAs detection [86].

HULC has been detected with higher frequency in the plasma of HCC patients as compared to healthy controls using qPCR, and higher HULC detection rates were observed in HBV-positive samples [87]. Higher plasmatic HULC levels correlated with Edmondson grade, indicating a progressive upregulation of HULC from well-differentiated HCC samples (Edmondson grades I–II) to undifferentiated lesions (Edmondson grades III–IV) [87]. These results are in line with the findings from a subsequent study demonstrating that circulating HULC and LINC00152 were significantly increased in plasma samples from HCC patients [88]. In addition to confirming the potential of HULC as a biomarker for HCC, a very recent study reported that among the eight serum lncRNAs studied as potential HCC biomarkers, a panel comprising serum LINC00152, UCA1, and AFP has the greatest ability to diagnose HCC [89]. Other lncRNAs have been suggested as candidate biomarkers for HCC diagnosis in combination with α-fetoprotein (AFP): LINC00635 and ENSG00000258332.1 have been detected in serum exosomes of HCC patients and associated with poor prognosis [90], whereas SNGH1 has been shown to improve the accuracy of early diagnosis of HCC [91]. A tissue microarray (TMA) study of 68 paired HCC/adjacent non-tumor tissues detected 23 lncRNAs in HBV-positive HCC patients and two of them, uc001ncr and AX800134, accurately diagnosed HBV-positive HCC using PCR [92]. Interestingly, AX800134 has also been reported to be upregulated in HBx-expressing HepG2 cells [93]. Furthermore, high expression levels of LRB1 were detected in a cohort of 326 HCC-related patients and positively associated with clinical features (AFP expression), large tumor sizes, and tumor stage [94].

In conclusion, lncRNAs appear to be interesting biomarker candidates for HCC detection, but well-designed studies enrolling a larger number of patients with well-defined etiologies and using standardized techniques are required to establish the diagnostic/prognostic value of ex-lncRNAs in (HBV-related) HCC. Of note, the specificity of ex-lncRNAs as HCC biomarkers has also to be carefully addressed, since ex-lncRNAS are frequently associated with various cancers [94,95,96,97].

In Figure 3 and Table 1, the lncRNAs involved in the development and progression of HBV-related HCCs are listed according to their activity in the nucleus or in the cytoplasm, the biological functions targeted, and their eventual detection in the extracellular fluids. In particular, the last column of the table highlights a role of the lncRNAs as biological markers and suggests a possible signature to better characterize the HCC in patients (state, prognosis, or drug resistance). 

## 6. LncRNAs as Targets for Therapy in Hepatocellular Carcinoma

The wealth of evidence in line with the oncogenic or tumor suppressor role of lncRNAs in hepatocarcinogenesis provides a rational for the development of strategies targeting lncRNAs to prevent and/or treat HCC [106,107]. 

RNA targeting technologies can be classified into two broad categories according to their post-binding mode of action: (a) occupancy only with the modulation of RNA function without promoting degradation (i.e., translational arrest or RNA processing/splicing) and (b) degradation of the targeted RNA through endogenous enzymes such as RNase H (short synthetic antisense oligonucleotides (ASO) or gapmers) or argonaute 2 (RNA interference (RNAi)) [106,108,109,110,111]. The first therapeutic applications of ASOs were suggested more than 40 years ago, and various strategies to enhance ASO delivery have been developed, including chemical modification as well as bioconjugation with lipids (i.e., lipid nanoparticles, LNPs), peptides or sugars (i.e., N-acetyl-galactosamine/GalNAc) [106,108,110,111]. Some antisense drugs have recently been approved, and several others are being investigated in late-stage clinical studies [108,110]. The majority of the approved antisense drugs have been developed for rare genetic diseases with limited or no therapeutic options: the RNaseH ASOs Mipomersen (Familial Hypercholesterolemia, FH), Inotersen (Hereditary Transthyretin Amyloidosis, hATTR), and Volanesorsen (Familial Chylomicronemia Syndrome, FCS) from Ionis Pharmaceuticals and the RNAi Patisiran (hATTR), Givosiran (Acute Hepatic Porphyrias, AHP), Lumasiran (Hyperoxaluria) and Inclisiran (primary hypercholesterolemia, both heterozygous familial and non-familial, and mixed dyslipidemias) from Alnylam Pharmaceuticals [106,108,110]. Antisense drugs in earlier stages of clinical development comprise all modes of action and target cardiovascular, metabolic, and endocrine disorders, cancers, neurological diseases, inflammatory and infectious diseases [106]. No antisense drugs for HCC treatment have reached the clinical stage. Few antisense drugs have been developed to target miRNAs involved in human diseases: Lademirsen/RG-012 targeting miR21 for kidney fibrosis in Alport syndrome patients, the miR29b mimic MRG-201 for cutaneous scleroderma, Miravisen and RG101 targeting miR-122, an miRNA required for HCV replication, in chronic hepatitis C patients [106]. ASO and RNAi targeting HBV transcripts are already actively developed to inhibit viral replication in chronic hepatitis B patients [106]. Antisense drugs targeting lncRNAs have not yet entered clinical research. The efficacy of ASO- and RNAi-based gene knockdown for lncRNAs is influenced by target intracellular localization, being nuclear lncRNAs better suppressed by ASOs and lncRNAs with dual nuclear and cytoplasmic localization suppressed by both ASOs and RNAi [112]. These results and the possibility to potentiate ASO deliver to the liver by their conjugation with GalNAc [108,110,111] may guide the development of antisense drugs to target lncRNAs in HCC patients. Progress in chemical modifications and safer and more effective delivery technologies together with the increasing recognition of the physiological and pathological roles of lncRNAs in the liver continue to drive the interest for the development of new oligonucleotides therapeutics for liver diseases and hepatocellular carcinoma [11,106].

## 7. Conclusions

Due to the pervasive transcription of the genome, the majority of cellular RNAs are not transcribed into proteins. lncRNAs represent the largest class of transcripts in human cells and are involved in major pathways governing multiple biological processes ranging from pluripotency to senescence and cancer development. lncRNAs also play an important role in acute and chronic viral infections. While some lncRNAs may be induced/repressed in order to contribute to the cellular antiviral response by restricting viral infection, viruses can also deregulate cellular lncRNA levels to promote their replication/spread. Furthermore, defined lncRNAs deregulated in the course of viral infection can contribute to disease pathogenesis, while the deregulation of others appears as a bystander effect of viral infection. 

With more than 250 million infected individuals globally, HBV is a major public health treat. While vaccines to prevent HBV infection have been available for decades, to date, there is still no cure for chronic hepatitis B that is a major cause of HCC, which is one of the most deadliest cancers worldwide. Despite recent improvements, therapeutic options for HCC remain limited. HCC is frequently diagnosed at a late stage when patients are no longer eligible for curative treatments. Systemic therapies including multikinase inhibitors and/or immune checkpoint inhibitors have been designed for patients with advanced disease [113,114], but poor treatment outcome and chemoresistance are frequently observed. Thus, novel therapeutic strategies against HBV infection as well as HCC are urgently needed. Numerous lncRNAs have been shown to be deregulated in the course of HBV infection and HCC development. Moreover, lncRNAs have also been reported to play a role in resistance both chemo- and immunotherapy [115]. Thus, a better understanding of the lncRNAs involved in hepatocarcinogenesis will not only expand our knowledge on this multifactorial cancer but also contribute to identify new therapeutic targets and diagnostic biomarkers for HCC. 

Technological developments have enabled researchers to gain more and more data regarding the nature and potential lncRNA targets in vitro, but the molecular mechanisms underlying the role of lncRNA in vivo remain poorly characterized. Nevertheless, descriptive studies can contribute to identify biomarkers for HCC diagnosis. Identification of mRNA–lncRNA co-expression networks by bioinformatic analysis of data from the cancer genome atlas (TCGA) consortium highlighted potential biomarkers for HCC [116]. Since lncRNA can be easily detected in plasma samples, lncRNAs such as HULC may represent valuable blood-based biomarkers for HCC diagnosis/prognosis. Furthermore, the identification of genes directly regulated by lncRNAs could contribute to uncovering new druggable targets.

To further understand how lncRNA contribute to HBV-related HCC in vivo, well-defined patient cohorts, relevant animal models, and standardized technologies are needed. Recent breakthroughs in the management of orphan diseases thanks to the approval of ASO drugs for defined human diseases and improvement of ASO-based therapeutics open perspectives for future strategies to prevent and treat HCC.

## Figures and Tables

**Figure 1 cancers-13-03115-f001:**
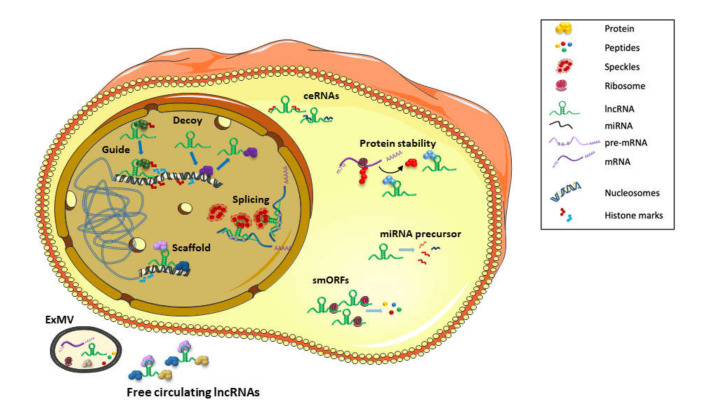
LncRNAs functions in HBV-related HCCs. The nuclear or cytoplasmic localization of lncRNAs reflects their mode of action. Nuclear lncRNAs are involved in nuclear organization, chromatin function by sequestering (decoy), guiding toward their target loci (guide) or tethering (scaffold) transcription factors or chromatin-modifying complexes and RNA maturation (splicing). In the cytoplasm, lncRNAs are involved in post-transcriptional mechanisms of gene regulation by: (a) binding to complementary sequences on miRNAs and relieving the repression of targeted mRNAs (Competing Endogenous RNA or ceRNAs); (b) binding to cytoplasmic proteins and affecting their stability; (c) acting as precursors of miRNAs; and (d) coding for small peptides through small open reading frames (smORFs). Circulating lncRNAs in extracellular fluids are found complexed to proteins (free circulating lncRNAs) or in extracellular microvescicles (exMVs), comprising both exosomes and larger MVs.

**Figure 2 cancers-13-03115-f002:**
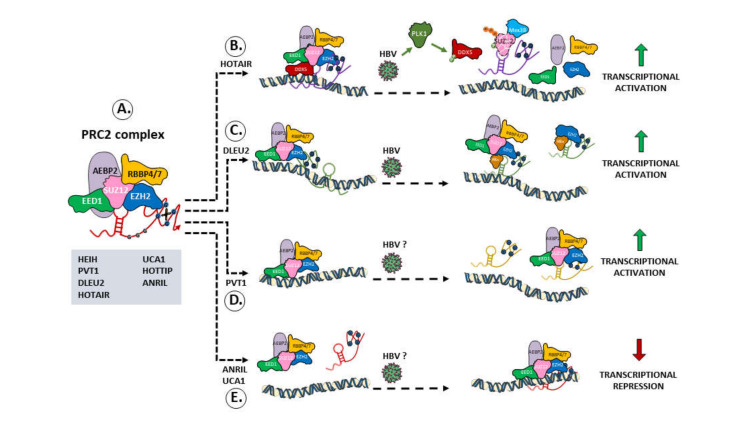
LncRNAs/PRC2 interactome in HBV-related HCCs. (**A**) Schematic representation of the PRC2 complex and its interaction with lncRNAs. The PRC2 complex proteins EED1, SUZ12, and EZH2 have a high affinity to G-quadruplex sequences, an intermediate affinity to unstructured G-rich RNA, and a lower affinity to stem-loop RNA structures on lncRNAs. The lncRNAs deregulated in HBV-related HCCs that bind the PRC2 complex are listed in (**A**). In (**B**) to (**E**) are shown the functional consequences of their interaction and HBV infection on the activity of the PRC2 complex. (**B**) The lncRNA HOTAIR interacts with the PRC2 complex and the RNA helicase DDX5. HBV infection promotes the activation of PLK1, which phosphorylates SUZ12 [36] and DDX5 [37]. The downregulation of DDX5 in HBV-infected cells and in HBV-related HCCs allows the preferential association of HOTAIR with the E3-ligase Mex3b, the ubiquitination of SUZ12 and its proteasomal degradation [37]. The functional significance of the phosphorylation of DDX5 by PLK1 remains to be determined but the transcriptional repression exerted by the DDX5/PRC2/HOTAIR complex is impaired, leading to the transcriptional activation of PRC2/HOTAIR target genes, including EpCAM and other genes related to the hepatic progenitor cells (HPCs) phenotype. (**C**) The lncRNA DLEU2, activated by HBV infection and upregulated in HCC, directly binds HBx and EZH2 [34]. By increasing DLEU2 levels and by binding to DLEU2, HBx interferes with EZH2/PRC2 repressive functions and activates transcription. HBx interaction with DLEU2 either evicts EZH2, a modality prevalent on cccDNA, or displaces the PRC2/EZH2 complex from close contact with chromatin in the case of host genes. PCR2 subunits are still immunoprecipitated with their target chromatin sequences [34]. The precise mechanisms underlying the removal of the repressive H3K27me3 mark imposed by the PRC2 complex, the deposition of active H3K27Ac marks, and the conversion of PRC2 complexes to transcriptional activation are still unclear. (**D**) The upregulation of the lncRNA PVT1 in HBV-related HCCs and its interaction with the PRC2 leads to a loss of EZH2 recruitment to the c-Myc oncogene promoter and the concomitant reduction of the H3K27me3 mark. Interestingly, this leads to cell proliferation and the acquisition of stem cell-like properties [51]. (**E**) The lncRNAs ANRIL and UCA1, both upregulated in HBV-related HCCs, guide PRC2 to the Kruppel-like factor 2 (KLF2) [52] and the P27/Kip1 [39] promoters, leading to an increase of H3K27 tri-methylation and their transcriptional repression.

**Figure 3 cancers-13-03115-f003:**
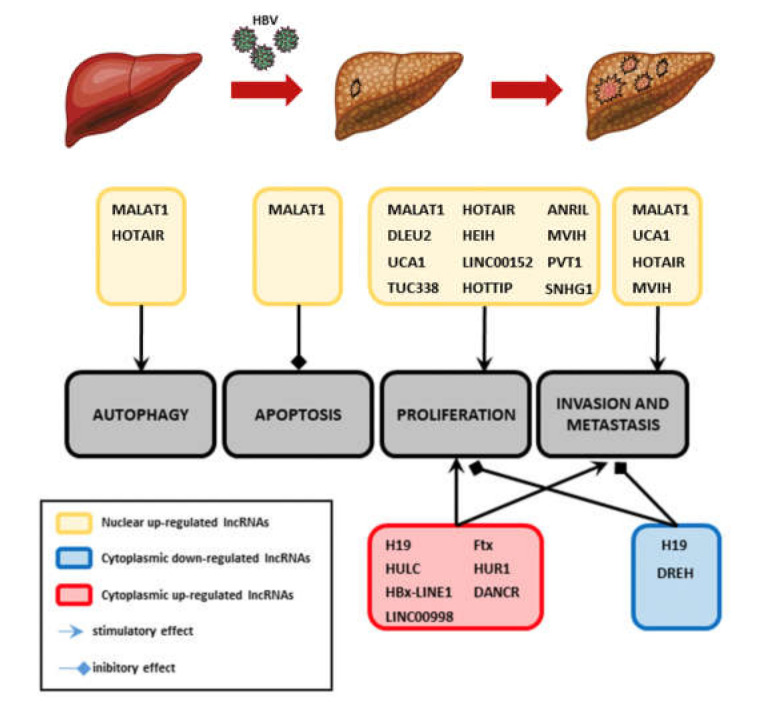
LncRNA involved in the development and progression of HBV-related HCCs. The biological functions targeted by lncRNAs deregulated in HBV-related HCCs and involved in HCC development and progression are shown. The lncRNA are grouped according to their up- or downregulation in the tumor and their activity in the nucleus or in the cytoplasm.

**Table 1 cancers-13-03115-t001:** LncRNAs deregulated in HBV-related HCCs.

LncRNA	Expression	Subcellular Localization	Biological Functions in HCCs	Detection in Extracellular Fluids
HEIH	UP	Nucleus	↑ Cell proliferation and tumor progression [32]	HCC [98]
HOTAIR	UP	Nucleus	↑ Tumor progression and poor prognosis (HBV-related HCCs) [35,36,37]	HCC [99]
UCA1	UP	Nucleus	↑ Cell proliferation and tumor progression [39,100]	HCC[89,96,99,101]
LINC00152	UP	Nucleus	↑ Cell proliferation [41] ↑ Tumor growth [40]	HCC [88,89]
TUC338	UP	Nucleus	↑ Cell proliferation [43,44]	
MALAT1	UP	Nucleus	↑ Tumor progression [45] Transformation of LPCs* [47,102]	HCC [89,99,103]
DLEU2	UP	Nucleus	Activation of cancer-related genes [34]	
PVT1	UP	Nucleus	↑ Cell proliferation and stem cell-like properties [49,51]	HCC [104]
ANRIL	UP	Nucleus	↑ Cell proliferation [52]	-
MVIH	UP	Nucleus	↑ Tumor growth and angiogenesis [53]	-
HOTTIP	UP	Nucleus	↑ Tumor progression and Poor survival (HCC) [54,55]	-
HULC	UP	Cytoplasm	↑ Tumorigenesis and tumor progression [60,61,62,63] Drug resistance [63,64]	HCC[87,88,89]
HBx-LINE1	UP	Cytoplasm	↑ Tumor growth [65,66]	-
HUR1	UP	Cytoplasm	↑ Cell proliferation and tumorigenesis [67]	-
DREH	DOWN	Cytoplasm	↓ Cell proliferation and tumor progression [68,69]	-
DANCR	UP	Cytoplasm	↑ Cell proliferation and stem cell-like properties [70,71]	-
Ftx	UP	Cytoplasm	↑ Tumor growth [75]	-
H19	UP / DOWN	Cytoplasm	↑↓ Tumor growth and progression [76,77] Drug resistance [78]	-
LINC00998	UP	Cytoplasm	↑ Cell proliferation, ↑ Tumorigenesis and Poor survival (HCC) [84]	-
Uc001ncr	UP	Cytoplasm	NR	HCC [92]
AX800134	UP	NR^1^	↑ Cell growth and invasion (HBx-expressing HepG2 cells) [93]	HCC [92]
SNHG1	UP	Nucleus	↑ Cell proliferation [105]	HCC [91]
LRB1	UP	NR	NR	HCC [94]

^*^ LPCs = Liver Progenitor Cells; ^1^ NR = not reported.

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
