# Peer review of "The lncRNAs in HBV-Related HCCs: Targeting Chromatin Dynamics and Beyond"

_cancers, 2021, doi:10.3390/cancers13133115_

Round 1
Reviewer 1 Report
This review article is talking about the lncRNAs in HBV-related HCCs. They summarize current knowledge on the role of lncRNAs in HBV infection and HBV-related liver carcinogenesis and discuss the potential of lncRNAs as predictive or diagnostic biomarkers. The results are well organized. However, some typos need to be checked. Minor Points: 1. In table 1, I was confused the “body fluids” column. I don’t know the reason the author set this column. Is the author trying to descript the lncRNA can be found in the body fluids? Please explain this column in detail. 2. Some references are cited in table 1 but did not found in the literature. Please add the information. 3. In table 1, please add the information of the “Biological functions in HCCs”. For example, HOTAIR upregulation promote or inhibit tumor progression.Author Response
Answer to Reviewer # 1
We are grateful to Reviewer 1 for defining our manuscript “well organized” and for giving us the opportunity to further improve the quality of our MS.
Regarding the minor points we acknowledge the concern raised by the Reviewer about the interpretation of Table 1:
- we agree with the Reviewer that “body fluids” is not the best heading to describe the content of that column. We have now replaced “body fluids” with “Detection in extracellular fluids”;
- we checked all references in the Table 1 (as well as in the whole text);
- we added, in the column "Biological functions in HCC", an arrow near each biological function to clearly indicate the role of each lncRNA in promoting or inhibiting a specific pathway.
- we added a sentence in the main text to better explain the content of Table 1.
Finally, we corrected several typos (that are now marked in red in the revised manuscript).
Reviewer 2 Report
Dear Editor,
I am really happy to make this review opinion about the 'Cancers-1209076', which was entitled with "The lncRNAs in HBV-related HCCs: targeting chromatin dynamics and beyond".
Vinvenzo et al, well described the HBV-related HCCs which are mainly regualted by lncRNAs. The present manuscript is quitely informative and provide imporatnt contents to understand pathophysiological features of HCC provoked by HBV. However, about the therapeutic access, still lack of information is available through the manuscript.
Therefore, after authors add more informative contents of therapeutics access it may be better to publish.
Thank you.
Author Response
We are grateful to Reviewer 2 for prompting us to expand the section on oligonucleotides therapeutics. We now describe concisely the different RNA-targeting technologies, the ASO and RNAi antisense drugs that have been approved for patients treatment and the challenges related to the development of antisense drugs targeting lncRNAs.
Reviewer 3 Report
Authors have put together a nice review on long-noncoding RNA (IncRNAs) in HBV-related HCC. This is a new area of cancer research. They have included many relevant studies that detected lncRNA in human samples. I enjoyed reading this. It is a good review, worth publishing. I have no negative comments. We need to know more about it. Thank you.
Author Response
We are pleased to learn that Reviewer 3 deemed our manuscript “good and worth publishing”.